

# Mesospheric temperature soundings with the new, daylight-capable IAP RMR lidar

Michael Gerding[1], Maren Kopp[1], Josef Höffner[1], Kathrin Baumgarten[1], and Franz-Josef Lübken[1]

[1]Leibniz-Institute of Atmospheric Physics at the Rostock University, Kühlungsborn, Germany

*Correspondence to:* Michael Gerding (gerding@iap-kborn.de)

**Abstract.** Temperature measurements by lidar are an important tool for the understanding of the mean state of the atmosphere as well as the propagation of gravity waves and thermal tides. Though, mesospheric lidar soundings are often limited to nighttime conditions (e.g., solar zenith angle >96°) due to the low signal-to-noise ratio during the day. By this, examination of long-period gravity waves and tides is inhibited, as well as soundings in summer at polar latitudes. We developed a new daylight-capable RMR lidar at our site in Kühlungsborn/Germany (54°N, 12°E) that is in routine operation since 2010 for temperature soundings up to 90 km or ∼75 km (night or day) and soundings of noctilucent clouds (NLC). Here we describe the setup of the system with special emphasis on the daylight suppression methods like spatial and spectral filtering. The narrowband Fabry-Pérot etalons for spectral filtering of the received signal induce an altitude dependant transmission of the detector. As a result, the signal is no longer proportional to the air density and the hydrostatic integration of the profile results in systematic temperature errors of up to 4 K. We demonstrate a correction method and the validity of correction by comparison with data obtained by our co-located, nighttime-only RMR-lidar where no etalon is installed. As a further example a time series of temperature profiles between 20 and 80 km is presented for day and night of 9/10 March 2014. Together with the other data of March 2014 these profiles are used to calculate tidal amplitudes. It is found that tidal amplitudes vary between ∼1 and 5 K depending on altitude.

## 1 Introduction

Temperature is one of the key parameters for the understanding of the atmosphere. The mean state is mainly a result of the residual circulation and the radiative budget, as well as the chemical and turbulent heating and cooling. Superposed, spatio-temporal temperature variations resulting from planetary, tidal and gravity waves are observed. Propagating waves become increasingly important in the middle atmosphere because of decreasing air density. Observational capabilities have improved in the last decade due to sophisticated satellite-borne instruments like TIMED/SABER, but several constraints remain like, e.g., a coarse spatio-temporal resolution. While lacking global coverage, ground-based lidars are the only technique for observation of temporal temperature variations on scales of 10–60 minutes in the middle atmosphere (e.g., Meriwether and Gerrard, 2004). However, lidars suffer from solar photon flux during the day, and therefore most lidar observations are limited to nighttime conditions. This hinders, e.g., measurements at polar latitudes in summer, the observation of tidal temperature variations, and



may also bias the retrieval of the seasonal temperature variation due to unresolved tides (Chen et al., 2000). Though, a potential bias might affect all data sets with limited temporal coverage like, e.g., sun-synchronous orbiting satellites.

Especially for resonance lidars a few daylight-capable systems have been developed, often using atomic vapor cells (e.g., Chen et al., 1996; Fricke-Begemann et al., 2002; Friedman et al., 2012) or Fabry-Pérot etalons (e.g., Chu et al., 2002) for
spectral filtering of the received signal. These lidars are typically limited to the $85 - 105$ km altitude range. Below, Rayleigh backscatter is used for temperature retrieval. Several systems cover the whole range up to $\sim 85$ km at night, but for daylight operation the range is typically limited to $\sim 60$ km (e.g., von Zahn et al., 2000; Klekociuk et al., 2003; Blum and Fricke, 2005) if available at all. While all these daylight-capable lidars are installed at polar latitudes, there is a general lack of temperature data during day from mid-latitudes, especially for the mesospheric altitudes above 60 km.

We have developed a daylight-capable Rayleigh-Mie-Raman (RMR) lidar at Kühlungsborn/Germany (54°N, 12°E) in 2009/2010. The new system has already proven to observe noctilucent clouds (polar mesospheric clouds) during day and night (Gerding et al., 2013b) as well as tidal temperature variations (Kopp et al., 2015). Soundings have been performed at solar elevations of up to 59°, i.e. the maximum elevation during summer. The new lidar is co-located with a nighttime-only RMR lidar being in operation since 1997 for observations of, e.g., temperatures (Alpers et al., 2004; Gerding et al., 2008;
Rauthe et al., 2008) and noctilucent clouds (Alpers et al., 2000; Gerding et al., 2013a).

Here we describe the instrumental setup of the new RMR lidar at Kühlungsborn (Section 2). We focus on the special requirements for daytime operation. Narrowband spectral filtering entails a somewhat different temperature retrieval, as the backscatter signal is no longer proportional to the air density (see Section 3). We describe the temperature retrieval for the new lidar and compare the results exemplarily with data simultaneously obtained with the old RMR lidar. In Section 4 we present
an example for temperature lidar soundings covering day and night. We also show tidal temperature variations for this period, affirming the results of Kopp et al. (2015). In some examples the data of the RMR lidar are complemented by observations with the co-located potassium resonance lidar (von Zahn and Höffner, 1996). The potassium lidar allows daytime soundings since year 2000 (e.g., Fricke-Begemann and Höffner, 2005).

## 2   Instrumental setup of the new IAP RMR lidar

We installed a new RMR lidar at Kühlungsborn in 2009/2010 and started regular operation in summer 2010. The design requirements for the RMR lidar are listed below.

- reduction of background count rate by about five orders of magnitude with a combination of spatial filtering (narrow field-of-view) and spectral filtering (narrowband detector)

- co-axial transmitter and receiver for complete overlap in the stratosphere and mesosphere

- high dynamic range of the detector system for signal coverage between $\sim 20$ and $90$ km

- detection of elastic ($532$ nm) and inelastic ($N_2$-Raman shifted, $608$ nm) backscatter for future aerosol correction in the lower stratosphere



- widely automated system (e.g., active beam stabilization), allowing regular observations even with briefly trained staff

- operation at broken clouds

The new lidar is optimized for soundings of middle atmospheric temperatures and noctilucent clouds (NLC) during both day and night. Details of the instrumental setup are shown in Figure 1 and Figure 2 .

The flashlamp pumped power laser Newport Spectra Physics Quanta-Ray PRO 290-30 is seeded by the continuous wave laser Innolight Prometheus 50NE (Fig. 1). The external seed laser allows a better wavelength control and wavelength monitoring compared to the internal seed laser. For this, it is locked to an iodine line by means of absorption spectroscopy using the frequency-doubled output of the seed laser. Until June 2015 the iodine line 1111 ($\lambda_{vac}$=532.241 nm) was used. Since then we use line 1109 at $\lambda_{vac}$=532.255 nm, corresponding to a seeding wavelength $\lambda_{seed,vac}$=1064.51 nm or $\lambda_{seed,air}$=1064.22 nm for

vacuum and air, respectively. The new seeding wavelength results in the advantage of a second harmonic "in air" ($\lambda_{air}$=532.11 nm) being identical with a Fraunhofer line in the solar spectrum, i.e. a ∼25% reduced solar background. The emission wavelength of the seed laser is monitored by a High Finesse WSU wavelength meter. Additional fiber ports of the seeding system are used for adjustment of the optical bench, for seeder power monitoring etc. Due to its low signal-to-noise ratio (SNR) we actually do not use the fundamental laser output at 1064 nm, but only the second harmonic at 532 nm. The frequency-doubled output of the

power laser is guided through a 10x beam widening telescope (BWT) for reduction of the beam divergence from <0.5 mrad to 0.05 mrad. Two steering mirrors and one fixed mirror are used to guide the light into the atmosphere, co-axially with the receiving telescope. The first of these steering mirrors is Piezo-mounted and used for beam stabilization (Eixmann et al., 2015). The computer-controlled beam stabilization fixes the beam axis to the optical axis of the telescope on a single pulse basis to overcome the effects of, e.g., atmospheric turbulence on the geometrical overlap of beam and telescope. The second steering

mirror is used for coarse adjustment of the beam axis, e.g., if the range of the Piezo mirror is too small due to slow but large thermal drifts. The coarse steering needs to be done only a few times per 24 h.

The backscattered light is collected by a 32" (∼81 cm) parabolic f/4 mirror (c.f. Table 1). The telescope mount is made from carbon-fiber reinforced plastic, ensuring minimal thermal expansion. As described by Eixmann et al. (2015), 90% of the received photons are guided to a fiber cable for lidar detection and 10% are used for beam stabilization. The fiber cable has a

25 diameter of only 0.2 mm, resulting in a field of view (FOV) of only 62 $\mu$rad. The small FOV has two major advantages: i) The number of photons received from scattered sunlight is much smaller compared to larger FOVs, e.g. a factor of ∼100 smaller compared to our nighttime-only RMR lidar. ii) The thin fiber with numerical aperture NA = 0.11 allows to build up a highly efficient detector with small divergence at small (∼ 22 mm) beamwidth, see below. The narrow FOV requires an active beam stabilization because otherwise thermal drifts and atmospheric turbulence would tilt the laser beam out of the FOV within

30 a few seconds. The optical setup of the receiver system is shown in Figure 2 . A chopper is used to prevent the detectors from the large number of photons scattered in the lower atmosphere. First, $N_2$ Raman shifted photons at 608 nm are guided to an Avalanche Photo Diode (APD). This detector is only operational during the night due to unsufficient SNR during the day. A narrowband interference filter (IF, FWHM = 130 pm) at 532 nm blocks a large part of the received sunlight. The whole setup up to the IF is shielded by hardboard to prevent multi-scatter and light leakage that may allow undesired photons hitting



the detectors. Three different detectors are used for the elastic backscatter signal at 532 nm. The combination of these allows to monitor the transmission of the etalons. Additionally, the lower sensitivity of detector 532/1 in combination with the 5% beamsplitter allows to measure in the lower stratosphere where the detector 532/3 would be saturated.

Further reduction of the solar scatter is achieved by a double Fabry-Pérot etalon, with technical details described in Table 1. In order to keep the setup simple and reliable, we use fixed-spacer etalons in a pressurized housing instead of Piezo-controlled etalons. Tuning of the etalons is done by changing the pressure inside the stainless steel housing, using either backscattered light from the atmosphere or light of the seed laser. The transmission of the etalons is ~92%. Photon counts are recorded by transient recorders (manufactured by Licel, Germany) and accumulated over 1000 laser pulses (~ 30 s). In an optional "cloud mode" the data are pre-integrated over 50 laser pulses and only used for final integration if the signal-to-noise ratio is sufficiently large. By this we can also operate during the gaps between fast travelling clouds that would otherwise be too short to be usable. Note that even a few seconds of "cloudy" data would contaminate the whole profile due to the strongly increased background. The vertical resolution of the profiles is 15 m. The signals are recorded up to 245 km altitude, where the range gates above ~117 km are used for background calculation.

Overall, the background count rate from solar backscatter is reduced by about five orders of magnitude compared to our nighttime RMR lidar. Figure 3 shows the diurnal change of the background count rates for the new, daylight-capable RMR lidar. The background count rate of the nighttime lidar is extrapolated from twilight conditions and the data of the new lidar, taking the different efforts of spectral and spatial filtering into account. The solar elevation is shown for comparison. The background count rate of the new lidar starts rising shortly before sunrise. It increases by ~2 orders of magnitude within about one hour until the sun is a few degree above the horizon. Later, the background increases slower until its maximum near solar culmination. For comparison, the background count rate of our old RMR lidar, representing a standard RMR lidar, increases already one hour earlier at solar elevation of $\sim -8°$. Typically, this lidar is switched of near 10 kHz background count rate due to insufficient signal-to-noise ratio. During nighttime, background count rate is still higher than with the new lidar due to higher dark count rate of the detector. As demonstrated in the next sections, effective background suppression with our new RMR lidar allows to observe mesospheric temperatures up to $\sim 75$ km during the day (~90 km during the night). Actually, temperature calculations are limited to altitudes above 30 km. In near future, the 608 nm channel shall be used for aerosol correction below $\sim 30$ km (c.f., e.g., Gerding et al., 2008). Furthermore, the rotational Raman channel of the old, nighttime RMR lidar, being equipped with a separate 50 cm telescope, shall be used with the new lidar during the night.

The new RMR lidar is operated semi-automatically and can be controlled by briefly trained staff. Depending on weather conditions more than 1000 h of data are obtained per year.

## 3 Temperature retrieval with transmission correction

Temperatures are calculated using the well-known Rayleigh integration method as described by Hauchecorne et al. (1991). This method relies on the fact that the observed backscatter profile is proportional to the air density profile. Unfortunately, for our new RMR lidar, because of the narrowband daylight filter, this is only true after the correction described below. As





described in Table 1, the bandwidth (FWHM) of the first etalon is ∼4.4 pm. This is in the range of the Doppler broadening of the backscattered radiation (∼2.5 pm, depending on temperature). Due to the Airy-function shaped transmission function, part of the backscattered photons are already blocked. Figure 4 (left) shows the spectral shapes of the transmission function of the first etalon and the Doppler-broadened backscatter before and after the first etalon, and after the second etalon. The bandwidth

of the pulse laser has to be taken into account. While we do not have a possibility for direct measurement of the pulse width, we can estimate a Lorentz-shaped spectrum with ∼45 fm FWHM from the laser pulse length (Heisenberg limit). In fact, we have also calculated the effect of larger bandwidths and found that the additional correction for, e.g., 85 fm FWHM is much below 0.1 K. The convolution of the Lorentz-shaped laser spectrum with the Gaussian function resulting from the Doppler effect is a Voigt function. In Fig. 4 we show the Doppler broadened backscatter spectrum for 300 K temperature and compare with the

pure Gaussian that can be expected if the laser bandwidth can be neglected. Additionally we show the spectra after the light passing the first and the second etalon. Obviously, the shape of the transmitted spectrum has changed and by this the number of transmitted photons is reduced compared to the incident light. Please note that the effect of the etalon on the transmitted light is largest at high temperatures like in Fig. 4. The effect of the laser bandwidth on the Voigt function is larger at low temperatures, when the Doppler effect gets smaller, i.e. the relative effect of the Lorentz function gets larger.

For an isothermal atmosphere the transmission of the etalons would be constant and the observed backscatter profile would still be proportional to the air density. For a real, non-isothermal atmosphere the proportionality between signal and air density is no longer valid. The transmission of the double etalon depends on the atmospheric temperature at the altitude of scattering: For atmospheric temperatures of, e.g., 160 K and 300 K the transmission changes between 0.86 and 0.79 (Fig. 4, right), and the backscatter profile will be distorted.

In order to calculate temperatures by the Rayleigh integration method the backscatter profile needs to be corrected for the changing etalon transmission. This is done in our retrieval by an iterative method. First, an approximate temperature is calculated ignoring the changing transmission and assuming the backscatter profile to be proportional to density. This approximated temperature profile is then used to calculate a corrected backscatter profile, dividing the count rate at each altitude bin by the transmission of the etalons for the temperature-broadened signal at this particular altitude. If done repetitively, for typical

conditions the correction in the second iteration is already much smaller than the statistical uncertainty (not shown). Therefore we use only one iteration. Figure 5 shows an example for a temperature profile before and after the transmission correction ($T_{new,uc}$ and $T_{new,c}$), respectively). Data are obtained during darkness, allowing comparison with the simultaneous, co-located profile measured by the nighttime RMR lidar ($T_{old}$). The hydrostatic integration is initialized at 85 km by the temperature observed simultaneously by our co-located potassium lidar ($T_K$). During top-down hydrostatic integration in the mesosphere,

the uncorrected temperature data $T_{new,uc}$ of the new RMR lidar increase generally slightly stronger than $T_{old}$. Just above the stratopause differences get smaller again. ECMWF data cannot be used for comparison at the model's sponge layer above 65 km, where the differences to both lidar data sets ($T_{old}$, $T_{new,uc}$) are up to 20 K. But below that altitude the data confirm the observation of the old RMR lidar, indicating that $T_{new,uc}$ is wrong. Also the corrected temperature profile $T_{new,c}$ is shown in Fig. 5. Within the statistical uncertainty it closely agrees with $T_{old}$. The lower Rayleigh channels of both RMR lidars are used



below ∼48 km. They also agree within the statistical uncertainty. A correction may not be applied to the data of the new RMR lidar as the receiver before the first etalon is used.

Differences between temperature profiles of the old and new RMR lidar are plotted on the right sight of Fig. 5. Near 80 km the uncorrected new profile shows slightly lower values ($T_{new,uc} - T_{old} < 0$), but this difference is still within the combined
statistical uncertainty. Below, the effect of temperature dependent etalon transmission starts to dominate. Subsequently, $T_{new,uc}$ is warm biased by ∼3 K above the stratopause. This bias is not visible with the corrected new profile ($T_{new,c} - T_{old}$). Here, an additional difference occurs around 67 km. Like the discrepancy below 80 km this may due to different (signal dependent) smoothing windows used for both lidars. Furthermore, the statistical uncertainty ($1\,\sigma$) of the photon count profiles should be taken into account, resulting in an temperature uncertainty of ∼2.5 K for both RMR lidars at 80 km altitude.

## 4   Initial observations during day and night

The daytime-capable RMR lidar at Kühlungsborn is operated continuously since March 2011 with some additional observations in June/July 2010. Temperature soundings are performed whenever weather conditions allow for at least 3 h of operation, independent of daylight conditions. Until March 2016 we collected about 6200 hours of data. A detailed study of the temperature data set is outside the scope of this paper. Kopp et al. (2015) published a study about the tidal temperature variation at our
site (using only the upper 532 nm channel and temperature variations), while Gerding et al. (2013b) focussed on the diurnal variation of Noctilucent Clouds. Here we present some temperature data of March 2014 as well as the tidal amplitudes of this month to demonstrate the capabilities of the new RMR lidar at Kühlungsborn.

Figure 6 shows the temperature profile observed on 9 March 2014 at 12–14 UT. Solar elevation was up to 28°. The temperature retrieval was initialized for the higher 532 nm channel at 85 km by the CIRA-86 climatology (Fleming et al., 1990). The
data of the high Rayleigh channel is taken at 44 km altitude to start the integration for the low 532 nm channel. Below 22 km altitude the signal is blocked by the chopper and no measurements are available here. The lowermost data show a temperature minimum at 23 km with temperatures as low as 200 K. A stratopause is hard to identify in this individual profile, because temperatures vary strongly with altitude between 40 km and 70 km. The vertical temperature gradient is down to -6 K/km in the lower mesosphere. Above 70 km, temperatures decrease smoothly, but the increasing uncertainty of up to 10 K should be
taken into account. Of course, the large temperature variability can be questioned, as it may be an artifact due to issues of, e.g., geometric overlap, detector saturation or additional aerosol scattering. For a co-axial system like the system described here, complete geometric overlap is limited to the range where the telescope is optimally focussed. The nearfield of our telescope extends up to the lower stratosphere, but we do not have any indications for de-focussing above ∼20 km. Geometric overlap may decrease with altitude if the laser is not properly adjusted to the optical axis of the telescope, but this would result in a
cold bias across a wide altitude range, but not in oscillations of temperature. Effects of detector saturation should be strongest at altitudes of highest signal, i.e. at 40–45 km, but not in the middle mesosphere where the signal is moderate. Aerosol layers might result in strongly varying temperatures if ignored in Rayleigh temperature retrieval. On the other hand, optically visible aerosol layers have not been reported above 50 km, so far. We have also plotted the ECMWF temperature profiles for 12 UT





and 0 UT (following day). Both demonstrate large variability similar to the lidar data, even if the model resolution is not high enough to reproduce our observations in full detail. Remarkably, below 40 km the 12 UT profile is in good agreement with our data, while the temperature minimum at 23 km is better reproduced by the 00 UT data.

During 9/10 March 2014 soundings continued for about 40 h. Figure 7 shows a subsection of 24 h. Only the deviations of the

5 individual profiles from the 24 h mean are plotted to highlight the temperature variations. Downward propagating structures of different periods and vertical wavelengths are visible and are presumably due to superposing gravity and tidal waves. Temperature variation depends on altitude and is up to ±15 K. As in Fig. 6 the retrieval for all profiles is initialized at 85 km by CIRA-86 data. Because the climatological temperature might deviate from the true temperature, we assume an uncertainty for the seed temperature of 10 K, adding to the uncertainty of photon statistics. We limit our retrieval to 10 K overall uncertainty,

resulting in a uppermost altitude of 78 km during night and 75 km during day. If different start altitudes for day and night are acceptable, temperature soundings can be extended during night up to at least 85–90 km by choosing higher start altitudes.

A single 24 h time series does not allow distinguishing between gravity waves and tides and, e.g., estimating tidal amplitudes. As reported by Kopp et al. (2015), about 100 h of data are necessary to get reliable fits of the diurnal tide and higher harmonics from temperature composites. Overall, in March 2014 we have 125 h of temperature data available (with ≥6 h/d). In this

15 month tidal amplitudes strongly vary with altitude (Figure 8 ). The diurnal tide maximizes at ∼43 km (amplitude ∼5 K), nearly vanishes below 50 km and generally increases with altitude in the mesosphere. A secondary maximum is found around 28 km. The semidiurnal tide in March 2014 is alternately increasing and decreasing, suggesting several filtering layers for the particular tidal mode. Amplitudes are between 0.5 and 3 K. The altitudinal structure of the terdiurnal tide is similar, but with a maximum at ∼67 km. In general, at altitudes above 50 km the higher harmonics often dominate against the diurnal tide

or are at least similar. Though, the maximum of the diurnal tide near 45 km confirms our previous observations. The March amplitudes below the stratopause are often around 5 K and, by this, much larger than during the rest of the year. Above 50 km we observe some variation in the March tidal amplitudes. Compared to the data published by Kopp et al. (2015), the terdiurnal tide was much stronger in 2014, while the semidiurnal tide was of similar strength. The comparison of the data presented here with previous publications demonstrates the necessity of continuous temperature soundings in the middle atmosphere. Our

temperature soundings during day and night allow for the retrieval of tides and their interannual, seasonal, and – if weather allows – short term variability.

## 5    Conclusions

We have demonstrated first temperature soundings by RMR lidar during full daylight (solar elevation much larger than 40°) in the middle atmosphere up to ∼75 km. Observations during daylight conditions request a special design of the lidar receiver that

effectively suppresses the background count rate from solar photons, but on the other hand has a high transmission at the laser wavelength. We have described our setup with a narrow telescope FOV (60 μrad instead of 600–1000 μrad of a standard RMR lidar) equipped with active beam stabilization. In addition to this spatial filtering we applied a narrowband interference filter and double Fabry-Pérot etalon (FPE) with 4.4 and 7.0 pm bandwidth (FWHM) for spectral filtering. Overall the solar background



is reduced by ∼5 orders of magnitude compared to a typical nighttime RMR lidar. A seed laser is frequency-locked to an iodine absorption line and assures a constant laser emission at the wavelength of the lidar detector. Making use of a Fraunhofer line in the solar spectrum the background is further reduced by 25% without any signal loss. The narrow FOV allows for a high transmission of the FPE of ∼92% at the laser wavelength. The narrowband FPE partly block the Doppler-broadened

Rayleigh backscatter, perishing the proportionality of the signal to the atmospheric density. Therefore the classical Rayleigh temperature retrieval can not be applied. We have described a data correction scheme that is used to overcome this issue. The validity of our retrieval is demonstrated by comparison with our old, nighttime RMR lidar and ECMWF operational data. The lidar system design is optimized for easy operation, allowing for more than 6200 h of data since summer 2010. Some examples for temperature soundings in the middle atmosphere during full daylight are presented. The capability for, e.g., observation of

tides is demonstrated, showing a dominating diurnal tide in the upper stratosphere in March. In summary, we have presented our approach of building an effective lidar for temperature soundings in an altitude region, where daylight soundings are sparse and, e.g., resonance lidars can not operate.

*Acknowledgements.* We thank our IAP colleagues Torsten Köpnick, Jens Fiedler, Jens Söder, and Hägar Holste for their particular contri-

butions to the new RMR lidar at Kühlungsborn. Michael Priester has been helpful in lidar operation and maintenance. This work has partly

been supported by the Deutsche Forschungsgemeinschaft (DFG) under grant GE1625/2-1.



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





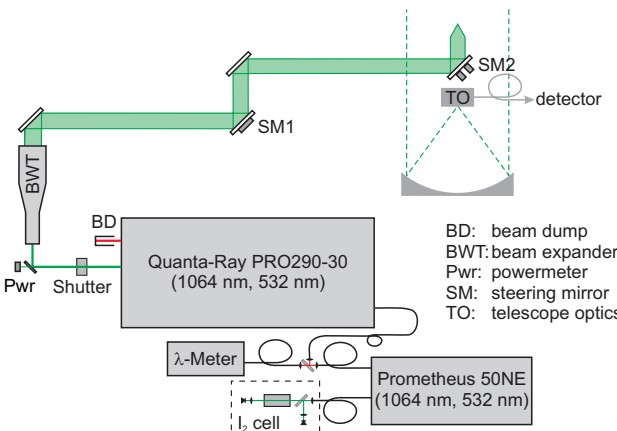

**Figure 1.** Setup of the transmitter of the new RMR lidar. Only the most important fiber cables of the seed laser are shown, additional fibers are available, e.g., for adjustment of the detector. SM1 is Piezo-mounted for fast beam stabilization, SM2 is motorized for coarse adjustment of the beam axis.

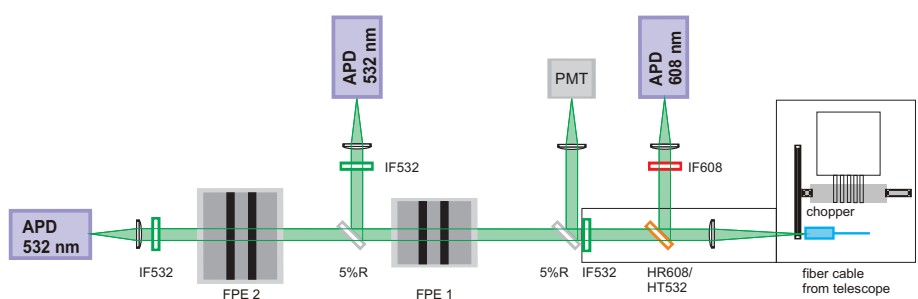

**Figure 2.** Setup of the receiver of the new RMR lidar. Photons collected by the telescope are guided to the receiver by a multimode fiber shown at the right side of the figure.

Meriwether, J. W. and Gerrard, A. J.: Mesosphere inversion layers and stratosphere temperature enhancements, Rev. Geophys., 42, RG3003, 2004.

Rauthe, M., Gerding, M., and Lübken, F.-J.: Seasonal changes in gravity wave activity measured by lidars at mid-latitudes, Atmos. Chem. Phys., 8, 6775–6787, 2008.

5  von Zahn, U. and Höffner, J.: Mesopause temperature profiling by potassium lidar, Geophys. Res. Lett., 23, 141–144, 1996.

von Zahn, U., von Cossart, G., Fiedler, J., Fricke, K. H., Nelke, G., Baumgarten, G., Rees, D., Hauchecorne, A., and Adolfsen, K.: The ALOMAR Rayleigh/Mie/Raman lidar: Objectives, configuration, and performance, Ann. Geophys., 18, 815–833, 2000.




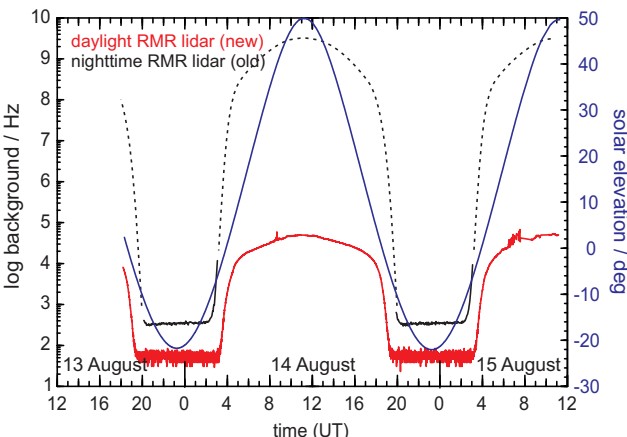

**Figure 3.** Temporal variation of the background count rate for the new RMR lidar under summer conditions (red). The nighttime lidar was additionally operated during darkness (black line), and the background is extrapolated for daylight conditions (black dashed line). Solar elevation is shown for comparison (blue line).

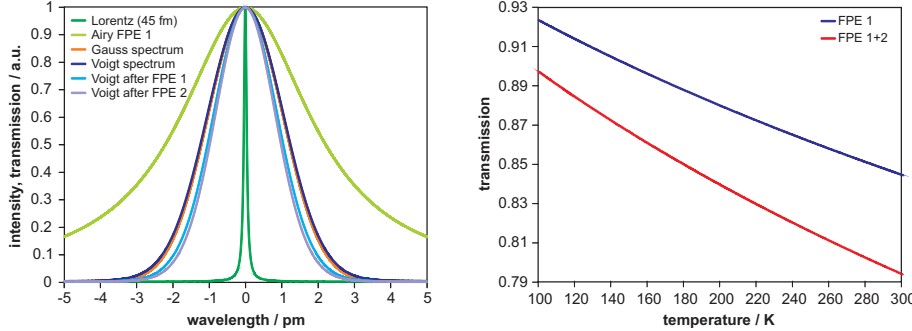

**Figure 4.** Left: Lorentz spectrum of the pulse laser (green), Airy transmission function of etalon 1 (light green) and Voigt-shaped, Doppler-broadened Rayleigh backscatter spectrum (for 300 K) before the etalons (blue), behind the first (light red) and second etalon (purple). A Gauss spectrum for 300 K is shown for comparison. Right: Transmission of the first and both etalons depending on atmospheric temperature, i.e. Doppler broadening.

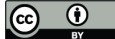



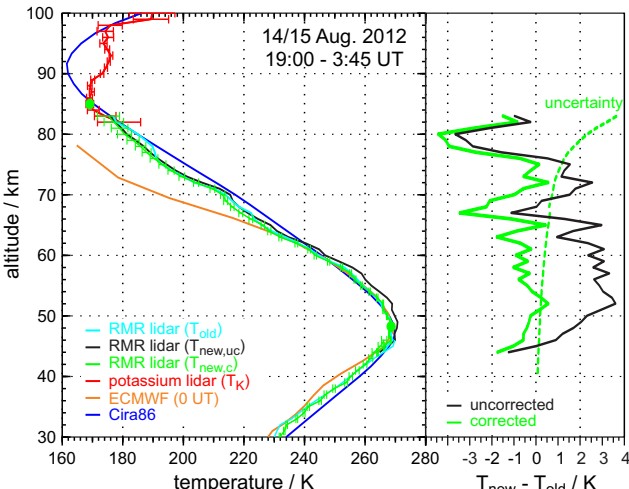

**Figure 5.** Nightly mean temperature profiles for 14/15 August 2012. Left: Data of the potassium lidar ($T_K$, red) is used for initialization of the hydrostatic temperature retrieval of the RMR lidars. Data of the old, nighttime RMR lidar $T_{old}$ shown in light blue, of the new RMR lidar in black ($T_{new,uc}$) and green ($T_{new,c}$). Data from CIRA-86 (blue) and ECMWF (orange) shown for comparison. Right: Differences between data of the old and new RMR lidar, before (black) and after (green) correction. The statistical uncertainty for the new RMR lidar is shown as dashed line and similar to the numbers for the old lidar.

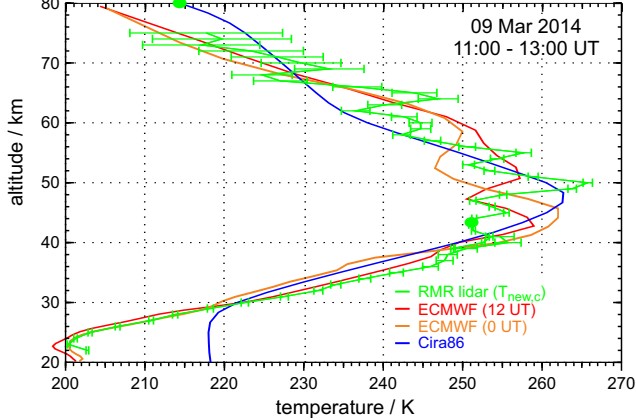

**Figure 6.** Temperature profile from the new RMR lidar (green, with error bars) during daylight for 9 March 2014. ECMWF data for 0 UT (orange) and 12 UT (red) as well as CIRA-86 data (blue) are shown for comparison.





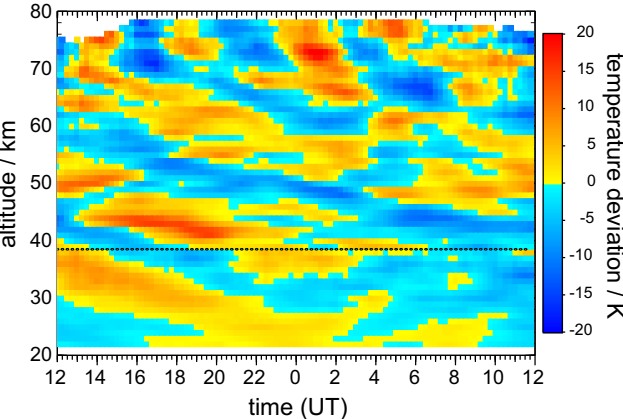

**Figure 7.** Time series of temperature data for 9/10 March 2014. Only deviation from the mean profile are shown. The black dotted line denotes the transition from the higher to the lower Rayleigh channel.

**Table 1.** Technical parameters of the IAP RMR lidar

| instrument parameter | value |
|---|---|
| wavelength (air) until June 2015, nm | 532.096 |
| wavelength (vac.) until June 2015, nm | 532.241 |
| wavelength (air) since June 2015, nm | 532.110 |
| wavelength (vac.) since June 2015, nm | 532.255 |
| repetition rate, pps | 30 |
| laser power @ 532 nm, W | $\sim$20 |
| beam diameter after expansion, mm | 90 |
| beam divergence after expansion, $\mu$rad | 50 |
| telescope diameter, m | 0.81 |
| telescope focal length, m | 3.25 |
| telescope field of view, $\mu$rad | 62 |
| interference filter width (FWHM), pm | 130 |
| etalon 1 free spectral range, pm | 120 |
| etalon 1 observed finesse | 27 |
| etalon 2 free spectral range, pm | 140 |
| etalon 2 observed finesse | 20 |




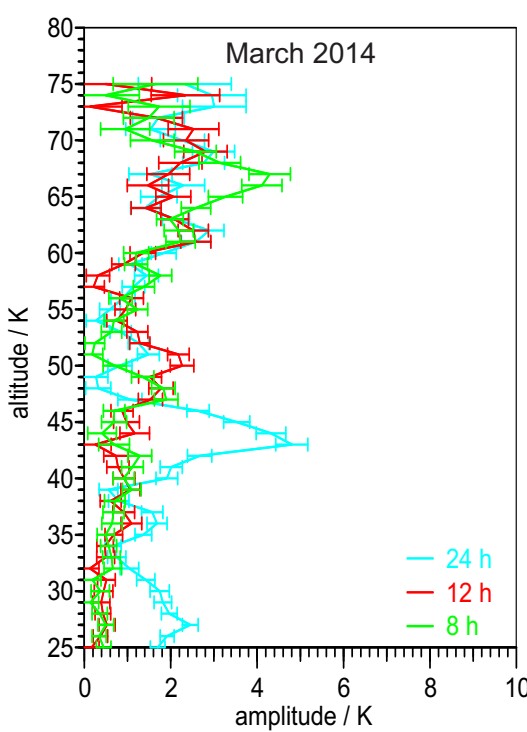

**Figure 8.** Amplitude profile for diurnal (blue), semidiurnal (red) and terdiurnal (green) tidal components from a composite of all March 2014 data.