# Peer review of "Mesospheric temperature soundings with the new, daylight-capable IAP RMR lidar"

_Atmospheric Measurement Techniques, 2016_

## Referee Comment (RC1) · Anonymous Referee #1 · 3 Jun 2016

Initial review of Mesospheric temperature soundings with new, daylight capable IAP RMR lidar by Gerding et al.

This paper provides a technical description and sample data and measurements of the daylight-capable Rayleigh-Mie-Raman (RMR) lidar at the Institute of Atmospheric Physics (IAP). This lidar system is a state-of-the-art instrument and the daytime measurement capabilities are a major contribution to the observation of the middle atmosphere. The instrument is not just significant because of it's measurement capabilities that allow observations over full diurnal cycles, but also because of it's stable operation that allows ready acquisition of observations (∼1000 h per year). The work is appropriate to Atmospheric Measurement Techniques and will be of interest to researchers.

The paper serves as an important technical companion paper to the recent paper by Kopp et al. (JASTP 2015) that presented measurements of the tides by the RMR lidar and highlighted the importance of full-day and multi-day operation in accurately characterizing the tides and their variability. I would like to see some of the technical and operational details expanded.

1) Can the authors add a raw data profile showing the signals in all four channels (3 Rayleigh, and 1 Raman)? A plot showing raw data profiles (that shows total signal including signal and background) representing observations over one hour at midnight and noon would be a valuable addition to the presentation.

2) Can the authors discuss the stability of the system in terms of the following key elements; a) Line center and line width of the laser transmitter. What is the accuracy and precision of the wave meter? Is the wave meter wavelength recorded on a per-shot basis? Is the b) Line canter of the pressure tuned Fabry-Perot etalons. What is the sensitivity of the line center to changes in pressure and temperature? How are the temperature and pressure monitored and maintained? c) Pointing jitter in the steering mirrors. How does the jitter compare with the 12 micro-radian margin between the receiver field-of-view (62 micro-radian) and transmitter beam divergence (50 micro-radian)? d) Do any of the variations (a-c) impact the narrowband filter correction, and if so can you characterize this uncertainty in the temperature retrievals?

3) The presentation of the narrowband filter correction is valuable (Figure 5). A third curve showing the difference between Tnew-uc and Tnew-c would be useful. The authors colds also cite maximum, minimum and typical differences in the uncompensated and compensated temperatures.

4) The presentation of the filter and line shape in Figure 4 might be clearer if the intensity and transmission were plotted on a logarithmic scale. Several of the curves are hard to distinguish.

5) The temporal resolution of the temperature measurements in Figure 7 is not re-

ported. From the pixels it appears to be about 15 minutes. Can the authors please cite the resolution of the measurement?

6) While comparison with ECMWF is interesting, is it possible to show a comparison with SABER?

7) Is Figure 7 the downward phase progressions appear to change phase speed above 70 km. The authors report tidal amplitudes unto 75 km. Do the authors feel that the change in phase above 70 km is geophysical in origin or perhaps reflects the initialization of the temperature retrievals at 85km?

---

## Referee Comment (RC2) · Anonymous Referee #2 · 7 Jun 2016

This paper provides a technical description of a state-of-the-art Raman-Mie-Rayleigh lidar system which is in operation at the Leibniz-Institute of Atmospheric Physics since 2010. Several techniques (small field of view, narrowband optical filters in the receiver, transmission at a Frauenhofer line) are used to reduce the solar background, thus making it possible to retrieve temperature profiles up to approximately 75 km in full daylight. This is a significant achievement as the new lidar system allows temperature observations in the stratosphere and lower mesosphere over full diurnal cycles. Such long observations are of scientific interest for studies of e.g. thermal tides, diurnal variation of (convective) gravity waves. The lidar system is also notable for its stable operation with more than 6000 hours of observations so far (> 1000 hours per

year). The work presented in this paper is appropriate for publication in Atmospheric Measurement Techniques.

Specific comments:

1. Gerding et al. present many interesting details of their lidar system in this paper. However, it is hard to compare the performance to other lidars based on retrieved temperature profiles. I suggest the authors add a figure showing a raw photon profile, e.g. one hour integration time.

2. The correction of spectral distortions in the backscatter signal induced by the double etalon is enlightening. I am wondering: did the authors consider validating their calculations by comparing signal ratios measured by detectors before and after the etalons for different altitudes (temperatures)? For example, the authors could derive the transmission of the double etalon at the stratopause (high temperature) and in the mesosphere (low temperature). The comparison could provide insight whether the instrument function of the (real) etalon is indeed an Airy type function.

3. The calculations are based on the assumption that the lidar transmits at the wavelength of peak transmission of the etalons. What is the precision of tuning the etalons to a specific wavelength? Can the authors provide an estimate of the temperature error caused by an improperly tuned etalon (e.g. wavelength of peak transmission is offset by 0.5 pm)?

Minor comments:

Page 3, line 11: "The emission wavelength of the seeder is monitored by a High Finesse WSU wavelength meter." Is the wavelength meter used to stabilize the seeder? Please clarify.

Page 3, line 24: "The fiber cable has a diameter of only 0.2 mm. . .". I assume the core of the fiber is 0.2 mm in diameter and the cable is larger.

Page 4, line 2 and Figure 2: Please mark detectors "532/1", "532/2", "532/3" in Figure

2.

Page 4, line 6: "Tuning of the etalons is done by changing the pressure inside the stainless steel housing". Please explain in more detail how the etalons are tuned. Is the transmission monitored as function of pressure? How often do the etalons need to be tuned?

Page 4, line 7: The transmission of the etalons ("∼92%") is very high. How was the transmission measured?

Page 4, line 14: "the background count rate form solar backscatter is reduced by about five orders of magnitude compared to our nighttime RMR lidar" How do the authors estimate the reduction in background count rate in daylight if the nighttime RMR lidar can only observe during darkness? Please explain. This also concerns Figure 3: How is the background extrapolated? Can the authors provide key parameters of the nighttime RMR system (e.g. field of view, bandwidth of interference filter)?

Page 5, line 1: "As described in Table 1…" The bandwidth of the etalons is not listed in Table 1.

Page 5, line 6: "… from the pulse length". Did the authors measure the pulse length? If yes, please provide information.

Page 5, line 7: "… calculated the effect of larger bandwidths and found that that the additional correction … is much below 0.1 K". How large is the initial correction for 45 fm bandwidth? How large is the wavelength jitter of the laser and how much does this jitter affect the transmission?

Page 5, line 18: "… the transmission changes between 0.86 and 0.79." Figure 4 (left) suggest that these numbers are valid for ideal etalons with 100% peak transmission. The transmission of the real double etalon would be lower in this case. Please clarify.

Page 6, line 7: "… this may be due to different (signal dependent) smoothing windows used for both lidars." Please explain.

Page 7, line 15: "The diurnal tide maximizes at ∼43 km (amplitude ∼5 K), nearly vanishes below 50 km . . ." "Vanishes" is not quite correct, in my opinion. The maximum (∼43 km) is below 50 km.

Page 7, line 17: "The semidiurnal tide in March 2014 is alternately increasing and decreasing, suggesting several filtering layers for the particular tidal mode". I am not entirely convinced. The vertical wavelength of the diurnal tide is large compared to the vertical separation of the "filtering layers" (e.g. minimum at 57 km, maximum at 62 km). Could the modulation in amplitude be caused by gravity wave-induced temperature perturbations which are not entirely suppressed in the composite analysis?

Figure 4: Caption reads ". . .Rayleigh backscatter spectrum before etalons (blue). . .", but the blue label reads "Voigt after FPE 1"

Figure 7: Please state the temporal resolution.

The language may be improved, e.g. "The computer-controlled beam stabilization fixes the beam axis. . .", "The thin fiber with numerical aperture NA= 0.11 allows to build up. . ."

---

## Author Comment (AC1) · 15 Jul 2016

**Author's response on "Mesospheric temperature soundings with the new, daylight-capable IAP RMR lidar" by Gerding et al.**

**Referee #1**

*Initial review of Mesospheric temperature soundings with new, daylight capable IAP RMR lidar by Gerding et al.*

*This paper provides a technical description and sample data and measurements of the daylight-capable Rayleigh-Mie-Raman (RMR) lidar at the Institute of Atmospheric Physics (IAP). This lidar system is a state-of-the-art instrument and the daytime measurement capabilities are a major contribution to the observation of the middle atmosphere. The instrument is not just significant because of it's measurement capabilities that allow observations over full diurnal cycles, but also because of it's stable operation that allows ready acquisition of observations (~1000 h per year). The work is appropriate to Atmospheric Measurement Techniques and will be of interest to researchers. The paper serves as an important technical companion paper to the recent paper by Kopp et al. (JASTP 2015) that presented measurements of the tides by the RMR lidar and highlighted the importance of full-day and multi-day operation in accurately characterizing the tides and their variability. I would like to see some of the technical and operational details expanded.*

We thank the reviewer for the helpful comments. Below we cite each comment (indicated by italics) followed by our answer. Line numbers are given with respect to the manuscript with marked changes.

*1) Can the authors add a raw data profile showing the signals in all four channels (3 Rayleigh, and 1 Raman)? A plot showing raw data profiles (that shows total signal including signal and background) representing observations over one hour at midnight and noon would be a valuable addition to the presentation.*

We have added a new Figure 4 in the revised manuscript, showing the raw data with and without background correction. We have chosen a one hour period near highest solar elevation from the same sounding also used in Figure 3 and 6 (old Figure 5), and a period 12 h later. Cf. page 4, lines 28-33.

*2) Can the authors discuss the stability of the system in terms of the following key elements; a) Line center and line width of the laser transmitter. What is the accuracy and precision of the wave meter? Is the wave meter wavelength recorded on a pershot basis?*

The seeder was stabilized to 532.096 nm (in air) until June 2015. Since then we use another iodine line for stabilization, resulting in a wavelength of 532.110 nm (cf. Table 1). The line center of the pulsed laser can actually not be measured. A so-called Laser Pulse Spectrometer is under construction, allowing such a measurement in future (cf. Baumgarten, AMT, 2008). Line width is (45 fm), estimated from pulse length as later described in Section 3. The Wave meter (30 MHz accuracy) is only used for coarse adjustment of the seed laser wavelength to the desired iodine line, while the fine control of the wavelength is done by iodine absorption spectroscopy. Wavelength stability of the seed laser is ~0.5 fm rms, i.e. much below the expected rms of the pulse laser. Wave meter and iodine measurements are

logged once per second. We have slightly changed the manuscript to make this clearer (new page 3, lines 12/13).

*b) Line center of the pressure tuned Fabry-Perot etalons. What is the sensitivity of the line center to changes in pressure and temperature? How are the temperature and pressure monitored and maintained?*

Pressure and temperature are not directly monitored, but the transmission of the etalons is measured continuously during the soundings. Room temperature is conditioned to 1 °C with insulation and large optical table as additional heat sink. The pressure is kept "constant" using a sealed housing. Transmission changes are below ~1% rms over a single day. Pressure is re-adjusted at max. ~10% transmission reduction of the double etalon, according to ~0.2 pm wavelength shift. This wavelength shift may result in 0.4 K over-correction of temperatures near the stratopause and less above and below. We have expanded the description of the double etalon and the discussion of the temperature correction accordingly (new page 4, lines 10/11 and page 6, lines 27-29).

*c) Pointing jitter in the steering mirrors. How does the jitter compare with the 12 micro-radian margin between the receiver field-of-view (62 micro-radian) and transmitter beam divergence (50 microradian)?*

The remaining pointing jitter is 3-5 microradian (cf. Eixmann et al, IEEE, 2015 and new page 3, line 20).

*d) Do any of the variations (a-c) impact the narrowband filter correction, and if so can you characterize this uncertainty in the temperature retrievals?*

We thank the reviewer for pinpointing some weaknesses in the manuscript. As described above in topic 2 c, we have added some sentences about potential temperature errors due to mis-adjustment of the etalons.

*3) The presentation of the narrowband filter correction is valuable (Figure 5). A third curve showing the difference between Tnew-uc and Tnew-c would be useful. The authors could also cite maximum, minimum and typical differences in the uncompensated and compensated temperatures.*

Many thanks for this suggestion. While we would like to avoid another curve for the sake of clarity of the figure, we will mention typical numbers for the correction instead (new page 6, lines 23-27).

*4) The presentation of the filter and line shape in Figure 4 might be clearer if the intensity and transmission were plotted on a logarithmic scale. Several of the curves are hard to distinguish.*

We follow this suggestion and show the left figure with logarithmic scale (new Figure 5).

*5) The temporal resolution of the temperature measurements in Figure 7 is not reported. From the pixels it appears to be about 15 minutes. Can the authors please cite the resolution of the measurement?*

We are sorry for this omission. The temperatures in this case are calculated every 15 min with 2 h integration (cf. new caption of Figure 8).

*6) While comparison with ECMWF is interesting, is it possible to show a comparison with SABER?*

We thank the reviewer for this suggestion. Instead of the ECMWF 00 UT profile we will show the coincident SABER profile (new Figure 7). SABER confirms the large temperature variation with altitude, even if the phase is somewhat different. This is not surprising, taking the spatial separation of lidar profile and SABER tangent point (900 km) into account. We will change the description of the figure accordingly (new page 7, lines 25-27).

*7) Is Figure 7 the downward phase progressions appear to change phase speed above 70 km. The authors report tidal amplitudes unto 75 km. Do the authors feel that the change in phase above 70 km is geophysical in origin or perhaps reflects the initialization of the temperature retrievals at 85km?*

We apologize for the misprint, but the temperature retrieval was initialized at 80 km. For our previous RMR lidar in combination with a K resonance lidar we have checked whether wave retrieval is influenced by the temperature seeding (true temperature or climatology). We found only little effects. Therefore we feel the wave parameters to be realistic up to 75 km.

---

## Author Comment (AC2) · 15 Jul 2016

**Author's response on "Mesospheric temperature soundings with the new, daylight-capable IAP RMR lidar" by Gerding et al.**

**Referee #2**

*This paper provides a technical description of a state-of-the-art Raman-Mie-Rayleigh lidar system which is in operation at the Leibniz-Institute of Atmospheric Physics since 2010. Several techniques (small field of view, narrowband optical filters in the receiver, transmission at a Frauenhofer line) are used to reduce the solar background, thus making it possible to retrieve temperature profiles up to approximately 75 km in full daylight. This is a significant achievement as the new lidar system allows temperature observations in the stratosphere and lower mesosphere over full diurnal cycles. Such long observations are of scientific interest for studies of e.g. thermal tides, diurnal variation of (convective) gravity waves. The lidar system is also notable for its stable operation with more than 6000 hours of observations so far (> 1000 hours per year). The work presented in this paper is appropriate for publication in Atmospheric Measurement Techniques.*

We thank the reviewer for the constructive comments. Below we cite each comment (indicated by italics) followed by our answer. Line numbers are given with respect to the manuscript with marked changes.

*Specific comments:*

*1. Gerding et al. present many interesting details of their lidar system in this paper. However, it is hard to compare the performance to other lidars based on retrieved temperature profiles. I suggest the authors add a figure showing a raw photon profile, e.g. one hour integration time.*

We have added a new Figure 4 in the revised manuscript, showing the raw data with and without background correction. Following a similar comment by Reviewer 1, we have chosen a one hour period near highest solar elevation from the same sounding also used in Figure 3 and 6 (old Figure 5), and a period 12 h later. Cf. page 4, lines 28-33.

*2. The correction of spectral distortions in the backscatter signal induced by the double etalon is enlightening. I am wondering: did the authors consider validating their calculations by comparing signal ratios measured by detectors before and after the etalons for different altitudes (temperatures)? For example, the authors could derive the transmission of the double etalon at the stratopause (high temperature) and in the mesosphere (low temperature). The comparison could provide insight whether the instrument function of the (real) etalon is indeed an Airy type function.*

The different channels needed for calculation of transmission have different signal strengths. The count rate between channel 532/1 and 532/3 varies by a factor ~100, i.e. the lowest channel covers the stratopause but not the upper mesosphere. Therefore such a test is unfortunately not possible. We have checked the validity of our assumptions by comparison with our old lidar for several different nights and the full available altitude range. A representative example is shown in this paper. Data in Table 1 are given for real illumination of the double etalon system. Finesse of etalon 2 is higher if illuminated by single-mode fiber without first etalon (ideal conditions).

*3. The calculations are based on the assumption that the lidar transmits at the wavelength of peak transmission of the etalons. What is the precision of tuning the etalons to a specific wavelength? Can the authors provide an estimate of the temperature error caused by an improperly tuned etalon (e.g. wavelength of peak transmission is offset by 0.5 pm)?*

We estimate the potential wavelength offset as maximal 0.2 pm. This offset would result in an over-compensation (i.e. underestimation of true temperature) by ~0.4 K. Please note that this potential error has a different sign than the effect of spectral broadening of the laser pulse. We thank the reviewer for this comment. We have improved the manuscript accordingly (new page 4, lines 10/11 and page 6, lines 27-29).

*Minor comments:*

*Page 3, line 11: "The emission wavelength of the seeder is monitored by a High Finesse WSU wavelength meter." Is the wavelength meter used to stabilize the seeder? Please clarify.*

As described in lines 7 and 12 of page 3, the seeder is locked to an iodine absorption line. The wavelength meter is only used for coarse adjustment and to write the approximate wavelength into a logfile.

*Page 3, line 24: "The fiber cable has a diameter of only 0.2 mm: : :". I assume the core of the fiber is 0.2 mm in diameter and the cable is larger.*

We write now "core diameter" (page 3, line 26).

*Page 4, line 2 and Figure 2: Please mark detectors "532/1", "532/2", "532/3" in Figure 2.*

Done. Thanks for the suggestion.

*Page 4, line 6: "Tuning of the etalons is done by changing the pressure inside the stainless steel housing". Please explain in more detail how the etalons are tuned. Is the transmission monitored as function of pressure? How often do the etalons need to be tuned?*

The pressure is changed continuously while tracking the transmission of the seeding light; first for 1st etalon, than 2 nd etalon. Transmission is controlled for every sounding comparing the different channels, and re-adjusted every 2-6 weeks. We have slightly extended the description (page 4, lines 10/11).

*Page 4, line 7: The transmission of the etalons ("_92%") is very high. How was the transmission measured?*

We measured the transmission with the seeder using collimated light from a single mode fiber. We have added the description, emphasizing that the number is valid for ideally collimated light (page 4, line 8).

*Page 4, line 14: "the background count rate form solar backscatter is reduced by about five orders of magnitude compared to our nighttime RMR lidar" How do the authors estimate the reduction in background count rate in daylight if the nighttime RMR lidar can only observe during darkness? Please explain. This also concerns Figure 3: How is the background extrapolated? Can the authors provide key parameters of the nighttime RMR system (e.g. field of view, bandwidth of interference filter)?*

We apologize for a partially misleading phrasing, because the effect of a new low-noise detector is included. New phrasing is "more than four orders of magnitude" (e.g., page 4, line 19). FOV is reduced from 800 microrad to 62 microrad (factor ~170); spectral width from 350 pm to ~4 pm (factor 88 if shape is neglected). Overall factor ~15000. The dashed line in Fig. 3 is only a rough estimation what can be expected for a "standard" nighttime system.

*Page 5, line 1: "As described in Table 1: : :" The bandwidth of the etalons is not listed in Table 1.*

We have changed the phrasing (page 5, lines 11/12).

*Page 5, line 6: ": : : from the pulse length". Did the authors measure the pulse length? If yes, please provide information.*

We measure the pulse length (10 ns, cf. page 5, line 17) by photodiode at least after each change of the flashlamps.

*Page 5, line 7: ": : : calculated the effect of larger bandwidths and found that that the additional correction : : : is much below 0.1 K". How large is the initial correction for 45 fm bandwidth? How large is the wavelength jitter of the laser and how much does this jitter affect the transmission?*

The correction is up to 3-4 K in the lower mesosphere, but depends on the actual temperature profile. The wavelength jitter of the pulse laser is unknown, but will be measured in future. From a similar laser (Baumgarten, AMT, 2008; Fiedler, ILRC 2008) we can estimate a jitter of 20-40 MHz, i.e. the effect is similar to a larger bandwidth of the laser. We have slightly changed the manuscript to make this clearer (page 6, lines 23-27).

*Page 5, line 18: ": : : the transmission changes between 0.86 and 0.79." Figure 4 (left) suggest that these numbers are valid for ideal etalons with 100% peak transmission. The transmission of the real double etalon would be lower in this case. Please clarify.*

The numbers are given with respect to an etalon with 100% transmission. We have improved the manuscript to make this clear (page 4, lines 29/30).

*Page 6, line 7: ": : : this may be due to different (signal dependent) smoothing windows used for both lidars." Please explain.*

The temperature profiles are smoothed by 1-2 km, depending on altitude. Further description can be found at Alpers et al., 2004 or Gerding et al., 2007. We have added the text accordingly (page 6, line 21).

*Page 7, line 15: "The diurnal tide maximizes at _43 km (amplitude _5 K), nearly vanishes below 50 km…" "Vanishes" is not quite correct, in my opinion. The maximum (_43 km) is below 50 km.*

We rephrase "gets small close to 50 km (< 0.5 K) and then generally ..." (page 8, line 6).

*Page 7, line 17: "The semidiurnal tide in March 2014 is alternately increasing and decreasing, suggesting several filtering layers for the particular tidal mode". I am not entirely convinced. The vertical wavelength of the diurnal tide is large compared to the vertical separation of the "filtering layers" (e.g. minimum at 57 km, maximum at 62 km). Could the modulation in amplitude be caused by gravity wave-induced temperature perturbations which are not entirely suppressed in the composite analysis?*

We rephrase "... suggesting several filtering layers in the course of the month for the particular tidal mode. Additionally it cannot be ruled out that some signatures of a strong gravity wave with about 12 h period are still visible even in the composite of 125 h of data." (page 8, line 8/9)

*Figure 4: Caption reads "… Rayleigh backscatter spectrum before etalons (blue)…", but the blue label reads "Voigt after FPE 1"*

We apologize for similarities in the line colors and an error in the caption. We improve the colors and change the caption accordingly.

*Figure 7: Please state the temporal resolution.*

We add "The profiles are calculated every 15 min with 2 h integration time." (new Figure 8)

*The language may be improved, e.g. "The computer-controlled beam stabilization fixes the beam axis…", "The thin fiber with numerical aperture NA= 0.11 allows to build up…"*

We edited the whole text again and improved several odd phrases.